# Step-DAD: Semi-Amortized Policy-Based Bayesian Experimental Design

**Marcel Hedman** [* 1]   **Desi R. Ivanova** [* 1]   **Cong Guan** [1]   **Tom Rainforth** [1]

## Abstract

We develop a semi-amortized, policy-based, approach to Bayesian experimental design (BED) called Stepwise Deep Adaptive Design (Step-DAD). Like existing, fully amortized, policy-based BED approaches, Step-DAD trains a design policy upfront before the experiment. However, rather than keeping this policy fixed, Step-DAD periodically updates it as data is gathered, refining it to the particular experimental instance. This test-time adaptation improves both the flexibility and the robustness of the design strategy compared with existing approaches. Empirically, Step-DAD consistently demonstrates superior decision-making and robustness compared with current state-of-the-art BED methods.

## 1. Introduction

Adaptive experimentation plays a crucial role in science and engineering: it enables targeted and efficient data acquisition by sequentially integrating information gathered from past experiment iterations into subsequent design decisions (Atkinson et al., 2007; MacKay, 1992; Myung et al., 2013). For example, consider an online survey that aims to infer individual preferences through personalized questions. By strategically tailoring future questions based on insights from past responses, the survey can rapidly hone in on relevant questions for each specific individual, enabling precise preference inference with fewer, more targeted questions.

Bayesian experimental design (BED) offers a principled framework for solving such optimal design problems (Chaloner and Verdinelli, 1995; Rainforth et al., 2024; Ryan et al., 2016). In BED, the quantity of interest (e.g. individual preferences), is represented as an unknown parameter $\theta$ and modelled probabilistically through a joint generative model on $\theta$ and experiment outcomes given de-

signs. The goal is then to choose designs that are maximally informative about $\theta$. Namely, we maximize the *Expected Information Gain* (EIG) (Lindley, 1956; 1972), which measures the expected reduction in our uncertainty about $\theta$ from running an experiment with a given design.

The *traditional* adaptive BED approach, illustrated in Fig 1a, involves iterating between making design decisions by optimizing the EIG of the next experiment step, and updating the underlying model through Bayesian updates that condition on the data obtained so far. Unfortunately, this approach leads to sub-optimal design decisions, as it is a greedy, myopic, strategy that fails to consider future experiment steps (Foster, 2021; Huan and Marzouk, 2016). Furthermore, it requires substantial computation to be undertaken at each experiment iteration, making it impractical for real-time applications (Rainforth et al., 2024).

Foster et al. (2021) showed that this traditional framework can be significantly improved upon by taking a *policy-based* approach (PB-BED). As shown in Fig 1b, their Deep Adaptive Design (DAD) framework, and its extensions (Blau et al., 2022; Ivanova et al., 2021; Lim et al., 2022), are based on learning a *design policy network* that maps from experimental histories to new designs. This policy is trained before the experiment, then deployed to make design decisions automatically at test time. This provides a *fully amortized* approach that eliminates the need for significant computation during the experiment itself, thereby enabling real-time, adaptive, and non-myopic design strategies that represent the current state-of-the-art in adaptive BED.

In principle, these fully amortized approaches can learn theoretically optimal design strategies (in terms of total EIG). In practice, learning a policy that remains optimal for all possible experiment realizations is rarely realistic. In particular, the dimensionality of experimental history expands as the experiment progresses, making it increasingly difficult to account for all possible eventualities through upfront training alone. Moreover, deficiencies in our model can mean that data observed in practice can be highly distinct from the simulated data used to train the policy.

To address these limitations, and allow utilisation of any available computation during the experiment, we introduce a hybrid, *semi-amortized*, PB-BED approach, called *Stepwise Deep Adaptive Design* (Step-DAD). As illustrated in

---

[*]Equal contribution   [1]Department of Statistics, University of Oxford.   Correspondence to: Marcel Hedman <marcel.hedman@stats.ox.ac.uk>, Desi R. Ivanova <desi.ivanova@stats.ox.ac.uk>.

*Proceedings of the $42^{nd}$ International Conference on Machine Learning*, Vancouver, Canada. PMLR 267, 2025. Copyright 2025 by the author(s).

Fig 1c, Step-DAD periodically updates the policy during the experiment. This allows the policy to be adapted using previously gathered data, refining it to maximize performance for the particular realization of the data that we observe. In turn, this allows Step-DAD to make more accurate design decisions and provides significant improvements in robustness to observing data that is dissimilar to that generated in the original policy training. Empirical evaluations reveal Step-DAD is able to provide significant improvements in state-of-the-art design performance, while using substantially less computation than the traditional BED approach.

## 2. Background

Guided by the principle of information maximization, Bayesian experimental design (BED, Lindley, 1956) is a model-based framework for designing optimal experiments. Given a model $p(\theta)p(y \mid \theta, \xi)$, describing the relationship between experimental outcomes $y$, controllable designs $\xi$ and unknown parameters of interest $\theta$, the goal is to select the experiment $\xi$ that maximizes the expected information gain (EIG) about $\theta$. The EIG, which is equivalent to the mutual information between $\theta$ and $y$, is the expected reduction in Shannon entropy from the prior to the posterior of $\theta$:

$$I(\xi, y) = \mathbb{E}_{p(y|\xi)}[H[p(\theta)] - H[p(\theta \mid \xi, y)]],$$

where $p(y \mid \xi) = \mathbb{E}_{p(\theta)}[p(y \mid \theta, \xi)]$ is the prior predictive distribution of our model.

### 2.1. Traditional Adaptive BED

BED becomes particularly powerful in adaptive contexts, where we allow the future design decision at time $t$, $\xi_t$, to be informed by the data acquired up to that point, $h_{t-1} := (\xi_1, y_1), \ldots, (\xi_{t-1}, y_{t-1})$, which we refer to as the *history*. In the traditional adaptive BED framework (Ryan et al., 2016), this is done by assimilating the data into the model by fitting the posterior $p(\theta \mid h_{t-1})$, followed by the maximization of the one-step ahead, or *incremental*, EIG

$$I^{h_{t-1}}(\xi_t) = \mathbb{E}\big[H[p(\theta \mid h_{t-1})] - H[p(\theta \mid h_t)]\big], \quad (1)$$

where the expectation is taken with respect to the marginal distribution $p(y \mid \xi_t, h_{t-1}) = \mathbb{E}_{p(\theta|h_{t-1})}[p(y \mid \theta, \xi_t, h_{t-1})]$. We use the superscript $h_{t-1}$ to emphasize conditioning on the history currently available, setting $h_0 = \varnothing$. This is a *closed-loop* approach (Foster, 2022; Huan and Marzouk, 2016), explicitly integrating all of the acquired data to refine beliefs about $\theta$ and inform subsequent design decisions.

Whilst this traditional framework offers a principled and systematic way to optimize experimental designs, it comes with some limitations. One drawback is its myopic nature that greedily maximizes for the next best design and overlooks the impact of future experiments, ultimately leading to sub-optimal design decisions. Another limitation is the

significant computational expense incurred from the iterative posterior inference and EIG optimization. In general, the posterior computation is intractable and the EIG (1) estimation is *doubly intractable* (Foster et al., 2019; Rainforth et al., 2018). Since both of these steps must be conducted at each step of the experiment, the traditional adaptive BED approach is often impractical for real-time applications.

### 2.2. Amortized Policy-Based BED

In response to the limitations of traditional adaptive BED, Foster et al. (2021) introduce the idea of amortizing the adaptive design process through learnt policies. This amortized policy-based BED (PB-BED) approach represents a significant advancement over the traditional framework, delivering state-of-the-art non-myopic design optimization whilst enabling real-time deployment.

PB-BED reformulates the design problem using a policy $\pi$, which maps experimental histories to subsequent design choices, $\pi : h_{t-1} \mapsto \xi_t$. The optimal policy is now the one that maximizes the *total* EIG across the entire sequence of $T$ experiments (Foster et al., 2021; Shen and Huan, 2021)

$$\mathcal{I}_{1 \to T}(\pi) = \mathbb{E}_{p(h_T|\pi)}\left[H[p(\theta)] - H[p(\theta \mid h_T)]\right] \quad (2)$$

$$= \mathbb{E}_{p(\theta)p(h_T|\theta,\pi)}\left[\log p(h_T \mid \theta, \pi) - \log p(h_T \mid \pi)\right] \quad (3)$$

where $p(h_T \mid \theta, \pi) = \prod_{t=1}^{T} p(y_t \mid \theta, \xi_t, h_{t-1}), p(h_T \mid \pi) = \mathbb{E}_{p(\theta)}[p(h_T \mid \theta, \pi)]$, and $\xi_t = \pi(h_{t-1})$ are all evaluated autoregressively. This policy-based formulation strictly generalizes the traditional adaptive BED approach, which can be viewed as using a specific policy that maximizes the incremental one-step-ahead EIG (1) at each iteration, that is $\pi_{\text{trad}}(h_{t-1}) = \arg \max_{\xi_t} I^{h_{t-1}}_{t-1 \to t}(\xi_t)$.

Whilst the total EIG formulation (2) provides a unified training objective for the policy, it remains doubly intractable like the standard EIG. The original Deep Adaptive Design (DAD) method (Foster et al., 2021) addressed this by using tractable variational lower bounds of the EIG (Foster et al., 2019; 2020; Kleinegesse and Gutmann, 2020) coupled with stochastic gradient ascent (SGA) schemes to directly train a policy network taking the form of a deep neural network directly mapping from histories to design decisions. It thus provided a foundation for conducting PB-BED in practice.

A number of extensions to the DAD approach have since been developed (Blau et al., 2022; Ivanova et al., 2021; Lim et al., 2022), broadening its applicability to a wider class of models by proposing alternative policy training schemes. All share a core methodology, where the policy is trained only once, offline, with experimental histories simulated from the model $p(\theta)p(h_T \mid \theta, \pi)$. Once trained, it remains unchanged during the live experiment and across multiple experimental instances (e.g. different survey participants), as illustrated in Fig 1b. This *fully amortized* approach eliminates the need for posterior inference and EIG optimization

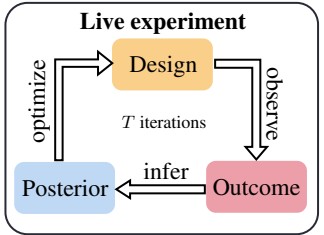

(a) Traditional adaptive BED

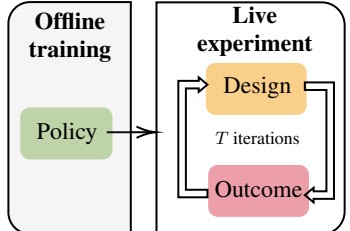

(b) Fully amortized PB-BED

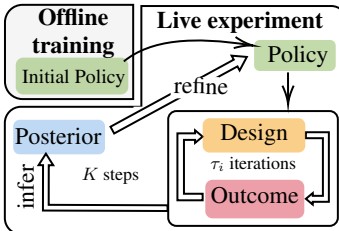

(c) Semi-amortized PB-BED

Figure 1: **Overview of adaptive BED approaches.** The traditional BED approach fits a posterior after each experiment iteration and optimizes for the next step best designs (i.e. greedily). Fully amortized policy-based BED approaches like DAD train a policy once offline, before the live experiment, then deploy this as a fixed policy to make adaptive design decisions during the experiment. Our semi-amortized approach enables periodic policy refinement during test-time.

at each step of the experiment, thereby allowing design decisions to be made almost instantly at deployment.

## 3. Semi-Amortized PB-BED

Fully amortized PB-BED methods enable real-time deployment and provide design decisions that are typically superior to those of the traditional framework. However, there are many problems where we can afford to perform some test-time training during the experiment itself. It is therefore natural to ask whether we can usefully exploit such computational availability to further improve the quality of our design decisions? In particular, the fact that the current state-of-the-art approaches for design quality are all fully amortized suggests that improvements should be possible when this is not a computational necessity.

To address this, we note that the computational gains of fully amortized PB-BED methods comes at the cost of their inability to adapt the *policy itself* in response to acquired experimental data. We argue that this rigidity leads to suboptimal designs decisions, particularly in scenarios where the real-world experimental data significantly deviates from the simulated histories used during training of the policy. Two primary factors contribute to this issue:

**Imperfect training** In fully amortized PB-BED we simulate experimental histories to try and learn a policy that will generalize across the entire experimental space—effectively learning a regressor from all possible histories to design decisions. However, the effectiveness of any learner with finite data/training is inevitably limited, especially in regions of the input space where training data is sparse. In short, we are learning a policy to cover all possible histories we might see, but at deployment we are dealing only with a specific history that may be similar to few, if any, of the histories we simulated during training. This challenge is particularly exacerbated in experiments with extended horizons, due to the high dimensionality of the resulting histories. Additionally, the finite representational capacity of the policy hinders perfect approximation even with in-

finite data. Together these lead to a discrepancy between the learned policy $\pi$ and the true optimal design strategy $\pi^*$, producing an *approximation gap* for the learned policies.

**Double reliance on the generative model** Fully amortized PB-BED relies on the generative model to both simulate experimental histories for policy training and to evaluate the success of our design decisions via the total resulting information gained. In other words, we use the model in both the expectation and information gain elements of the EIG in (2). This dual reliance magnifies the consequences of model misspecifications (Go and Isaac, 2022; Overstall and McGree, 2022). Moreover, even if the model is well-specified from a Bayesian inference perspective, there might still be significant discrepancies between the prior-predictive distribution, $p(h_T|\pi)$, used to simulate data in the policy training and the true underlying data generating distribution.

The upshot of this is that we may see data at deployment that is highly distinct from any simulated during the policy training. The lack of mechanisms for integrating real experimental data means that fully amortized approaches have no mechanism to overcome this issue. This can be characterized as a form of *generalization gap*—the learned policy fails to generalize to the real-world experimental conditions, due to its inability to integrate and respond to the actual experimental data gathered so far (Hastie et al., 2009).

### 3.1. Online policy updating

To address these limitations, we propose a *semi-amortized* PB-BED framework, which introduces dynamic adaptability by allowing periodic updates to the policy during deployment in response to acquired experimental data. In short, it will update the original policy at one or more points during the experiment, refining it to maximize the EIG of the remaining steps, conditioned on the data gathered so far.

The motivation behind this semi-amortized framework is the intuition that while a fully amortized policy is a strong starting point, it can be significantly enhanced through targeted refinements leveraging gathered data. Focusing for

now on the case of a single policy update, the following proposition formalizes this intuition and lays the theoretical foundation for semi-amortized PB-BED.

**Proposition 3.1** (Decomposition of total EIG). *For any design policy $\pi$, the total EIG of a $T$-step experiment can be decomposed as*

$$\mathcal{I}_{1 \to T}(\pi) = \mathcal{I}_{1 \to \tau}(\pi) + \mathbb{E}_{p(h_\tau \,|\, \pi)}[\mathcal{I}^{h_\tau}_{\tau+1 \to T}(\pi)], \quad (4)$$

*for any intermediate step $1 \le \tau \le T$, where*

$$\mathcal{I}^{h_\tau}_{\tau+1 \to T}(\pi) =$$
$$\mathbb{E}_{p(\theta|h_\tau)p(h_{\tau+1:T}\,|\,h_\tau,\theta,\pi)}\left[\log \frac{p(h_{\tau+1:T}\,|\,h_\tau,\theta,\pi)}{p(h_{\tau+1:T}\,|\,h_\tau,\pi)}\right]. \quad (5)$$

*Proof.* We can write the likelihood and marginal as

$$p(h_T\,|\,\theta,\pi) = p(h_\tau\,|\,\theta,\pi)p(h_{\tau+1:T}\,|\,h_\tau,\theta,\pi)$$
$$p(h_T\,|\,\pi) = p(h_\tau\,|\,\pi)p(h_{\tau+1:T}\,|\,h_\tau,\pi).$$

Substituting in (2) and rearranging now yields

$$\mathcal{I}_{1 \to T}(\pi) = \mathbb{E}_{p(\theta)p(h_\tau|\theta,\pi)}\left[\log \frac{p(h_\tau\,|\,\theta,\pi)}{p(h_\tau\,|\,\pi)}\right] +$$
$$\mathbb{E}_{p(h_\tau|\pi)p(\theta|h_\tau)p(h_{\tau+1:T}|h_\tau,\theta,\pi)}\left[\log \frac{p(h_{\tau+1:T}\,|\,h_\tau,\theta,\pi)}{p(h_{\tau+1:T}\,|\,h_\tau,\pi)}\right]$$
$$= \mathcal{I}_{1 \to \tau}(\pi) + \mathbb{E}_{p(h_\tau\,|\,\pi)}[\mathcal{I}^{h_\tau}_{\tau+1 \to T}(\pi)] \text{ as required.} \qquad \square$$

This decomposition of the total EIG into two distinct components—the EIG accumulated up to an intermediate step $\tau$, and the expected EIG for subsequent steps conditional on the history at that point $h_\tau$—demonstrates that the optimality of a policy for the latter phases of the experiment, from $\tau + 1$ to $T$, is solely determined by the model $h_\tau$.

To see this, first note that, without loss of generality, we can break down the definition of our policy into how it behaves when given histories of length less than $\tau$ and when it is given longer histories, such that $\pi(h_t) = \pi_0(h_t)$ if $t < \tau$ and $\pi(h_t) = \pi_\tau(h_t)$ if $t \ge \tau$. We can thus rewrite (4) as

$$\mathcal{I}_{1 \to T}(\pi) = \mathcal{I}_{1 \to \tau}(\pi_0) + \mathbb{E}_{p(h_\tau\,|\,\pi_0)}[\mathcal{I}^{h_\tau}_{\tau+1 \to T}(\pi_\tau)].$$

Here the first term is independent of $\pi_\tau$, while $\pi_0$ affects the second one only through its influence on the distribution of $h_\tau$. At deployment time, we will have a specific $h_\tau$ once we reach step $\tau$ of the experiment, and so the conditional optimal policy for the remaining steps given the data gathered so far is $\arg\max_{\pi_\tau} \mathcal{I}^{h_\tau}_{\tau+1 \to T}(\pi_\tau)$, which is independent of $\pi_0$.

Our semi-amortized framework is now based around exploiting this independence to refine the policy midway through the experiment by introducing a *step design policy* $\pi^s$. Initially, $\pi^s$ uses the fully amortized policy $\pi_0$ for the first $\tau$ steps of the experiment. Here $\pi_0$ trained as if would be used for the full experiment, such that we maximize its total EIG, $\mathcal{I}_{1 \to T}(\pi_0)$. After step $\tau$, $\pi^s$ switches to a new policy

$\pi_\tau$, which is trained to maximize the total *remaining* EIG, $\mathcal{I}^{h_\tau}_{\tau+1 \to T}(\pi_\tau)$, as defined in (5).

This gives us an **infer-refine** process for semi-amortization in PB-BED that mirrors the two stage procedure characteristic of traditional adaptive BED (cf Fig. 1a and Fig. 1c). The **infer** stage entails fitting the posterior distribution $p(\theta\,|\,h_\tau)$ with the data up to $\tau$. The subsequent **refine** stage learns a customized policy $\pi_\tau$ for the remaining steps of the experiment by maximizing (5). It therefore allows for more effective design decisions than the fully amortized approach. However, unlike the traditional BED approach, which is greedy and requires updates at every experimental step, our semi-amortized method offers a superior non-myopic design strategy and allows for selective updates.

It is important to acknowledge that this approach requires some computation during the experiment, which can pose challenges in applications where design decisions must be made very quickly. However, in many cases there is some computation time available, and our semi-amortized approach can exploit this, even if the available time is limited. In particular, as we will show in subsequent sections, improvements to the policy can often be achieved with minimal additional training, such that substantial gains are often possible without drastically compromising deployment speed. As such, semi-amortized PB-BED still maintains large computational benefits over traditional adaptive BED.

**Multi-step policy updates** We can naturally extend our approach to include a multi-step update mechanism, noting that Proposition 3.1 can be applied recursively to break down the total EIG into more segments. To this end, we define a *refinement schedule*, $\mathcal{T} = \tau_0, \tau_1, \cdots, \tau_K$—an increasing sequence defining the points at which the policy is refined. We adopt the convention $\tau_0 = 0$ and $h_0 = \varnothing$, marking the offline optimization of the fully-amortized policy $\pi_0$. For $\tau_k > 0$, we follow our two-stage infer-refine procedure. In general, the more often we refine the policy, the better it will be (albeit with diminishing returns), at the cost of increasing the required deployment-time computation.

## 4. Stepwise Deep Adaptive Design

We introduce **Stepwise Deep Adaptive Design** (Step-DAD) to implement our semi-amortized PB-BED framework in practice. Building on DAD and the infer-refine procedure outlined in the last section, Step-DAD employs stochastic gradient ascent schemes to optimize variational lower bounds on the remaining EIG (5) to sequentially train the step policy $\pi^s$ in a scalable manner. An overview of Step-DAD is presented in Algorithm 1 in Appendix A

The two key components of Step-DAD's aforementioned infer-refine procedure are an inference method for approximating $p(\theta|h_\tau)$, and a refinement strategy for using this to update our policy. Standard inference techniques (such as

variational inference or Monte Carlo methods) can be used for the former as discussed in our experiments. Our focus here will therefore instead be on our specialized procedure for policy refinement and the policy architecture itself.

### 4.1. Policy refinement

Due to its doubly intractable nature, the task of optimizing the remaining EIG, $\mathcal{I}_{\tau_k+1\rightarrow T}^{h_{\tau_k}}(\pi)$, presents a notable challenge. In selecting an appropriate scalable and efficient estimator for it, we wish to ensure compatibility with a wide range of inference schemes for $p(\theta \,|\, h_{\tau_k})$. Namely, as this serves as an updated 'prior' during the policy refinement, it is important that we use an EIG estimator that does not require evaluations of the prior density, to ensure compatibility with sample-based inference schemes.

Lower bound estimators such as the explicit-likelihood-based sequential Prior Contrastive Estimator (sPCE, Foster et al., 2021), as well as the implicit likelihood InfoNCE (Ivanova et al., 2021; van den Oord et al., 2018) and NWJ (Kleinegesse and Gutmann, 2020; Nguyen et al., 2010) bounds, align with this requirement. For generative models with explicit likelihoods (implicit models are discussed in Appendix B) we therefore use the sPCE bound:

$$\mathcal{L}_{\tau_k+1\rightarrow T}^{h_{\tau_k}}(\pi) = \mathbb{E}\left[\log \frac{p(h_{\tau_k+1:T}|\theta_0, h_{\tau_k}, \pi)}{\frac{1}{L+1}\sum_{\ell=0}^{L} p(h_{\tau_k+1:T}|\theta_\ell, h_{\tau_k}, \pi)}\right]. \tag{6}$$

Step-DAD parameterizes $\pi$ by a neural network and optimizes an appropriate objective, such as (6), with respect to the network parameters using stochastic gradient ascent (SGA) schemes (Kingma and Ba, 2014; Robbins and Monro, 1951). Following Foster et al. (2021), we use path-wise gradients in the case of reparametrizable distributions (Mohamed et al., 2020; Rezende et al., 2014), and score function (REINFORCE) otherwise (Williams, 1992).

### 4.2. Policy architecture

Similar to Foster et al. (2021), our policy architecture is based on individually embedding each design-outcome pair $(\xi_i, y_i) \in h_t$ into a fixed-dimensional representation, before aggregating them across $t$ to produce a summary vector. This allows for condensing varied-length experimental histories into a consistent dimensionality, to handle variable history sizes. The summary vector is then mapped to the next experimental design $\xi_{t+1}$. For the aggregation mechanism, the choice between permutation invariant and autoregressive architectures depends on the nature of the data. When the data $h_t$ is exchangeable, permutation invariant architectures like DeepSets (Zaheer et al., 2017) or SetTransformer (Lee et al., 2019) are suitable. In contrast, sequential or time-series data would benefit from autoregressive models like transformers (Vaswani et al., 2017).

In principle, one could train an entirely new policy $\pi_{\tau_k}$ at each refinement step $\tau_k$, potentially even varying the specific architecture between these. Though such a strategy may occasionally be advantageous, we instead, propose a more pragmatic and lightweight approach: leveraging the already established fully amortized policy $\pi_0$ as a baseline and fine-tuning it for subsequent steps. In our experiments we do this using full fine-tuning of all policy parameters, but one could instead implement more parameter-efficient methods if needed, for example, only adjusting the last few layers.

## 5. Related Work

The idea of using a design policy in the context of adaptive BED was first proposed by Huan and Marzouk (2016). Leveraging dynamic programming principles, the policy they learn aims to establish a mapping from explicit posterior representations—serving as the *state* in reinforcement learning (RL) terminology—to subsequent design choices. As a result, each iteration of the experiment necessitates substantial computational resources for updating the posterior. The concept of fully amortized policy-based BED, which directly maps data collected to design decisions, has only recently been introduced (Foster et al., 2021) and subsequently extended to differentiable implicit models (Ivanova et al., 2021) and downstream tasks (Huang et al., 2024). While Step-DAD uses the policy training approach of DAD and iDAD based on direct SGA of variational bounds (e.g. (6)), our semi-amortized PB-BED framework is also compatible with more RL-based design policy training approaches, like those of Blau et al. (2022) and Lim et al. (2022), which are more suited to discrete design spaces. We emphasize that none of these previous approaches have looked to refine the policy during the experiment itself.

As discussed in §2.1, adaptive BED has traditionally employed a two-step greedy strategy, involving posterior inference followed by an EIG optimization (Foster et al., 2019; 2020; Huan and Marzouk, 2014; Kleinegesse and Gutmann, 2019; 2020; 2021; Kleinegesse et al., 2021; Myung et al., 2013; Overstall and McGree, 2020; Price et al., 2018; Ryan et al., 2016; Vincent and Rainforth, 2017). While Step-DAD diverges from these in its EIG optimization by training policies instead of designs, it does share their need to perform posterior inference. The inference scheme used by previous work has varied between problems and the needs of the underlying Bayesian model being used, with sequential Monte Carlo (Del Moral et al., 2006; Drovandi et al., 2014; Rainforth, 2017; Vincent and Rainforth, 2017) and likelihood-free (Huan and Marzouk, 2013; Lintusaari et al., 2017; Sisson et al., 2018; Thomas et al., 2016) inference schemes proving popular. There is also growing recent interest in approaches that utilize inference itself as part of the EIG optimization. (Amzal et al., 2006; Iollo et al., 2024; 2025;

Iqbal et al., 2024a;b). Important considerations in choosing this scheme include the availability of an explicit likelihood, the ability to take derivatives, and computational budget.

The challenge of *model misspecification* in BED remains a critical, but relatively underexplored, problem (Farquhar et al., 2021; Feng et al., 2015; Go and Isaac, 2022; Overstall and McGree, 2022; Rainforth et al., 2024; Sloman et al., 2022). Fully amortized PB-BED is particularly vulnerable to model misspecification due to its reliance on a singular learning phase without the capacity to integrate real-world experimental feedback. As we will see in the experiments, our semi-amortized PB-BED methodology, whilst not directly tackling the issue of misspecification, typically does enhance robustness to misspecification, due to enabling iterative data integration and policy refinement.

Finally, BED shares important connections to reinforcement learning (Sutton and Barto, 2018). Most notably, it has been shown that the adaptive BED problem can be formulated as various forms of Markov Design Processes (MDPs (Doshi-Velez and Konidaris, 2016; Guez et al., 2012; Ross et al., 2007)), using the incremental EIG as the reward and either the posterior (Huan and Marzouk, 2016) or, more practically, the history as the state (Blau et al., 2022; Foster, 2021). Here PB-BED approaches are most closely linked with offline model-based RL (Kidambi et al., 2020; Levine et al., 2020; Moerland et al., 2023; Ross and Bagnell, 2012; Yu et al., 2020), in that they learn a policy upfront which can then be deployed. However, PB-BED varies in many significant ways from typical RL settings. For example, we *do not have access to any data to train our policy*, but are instead focused on optimal sequential decision-making under a given model. We can also typically directly use SGA to train the policy as we have access to end-to-end differentiable objectives and our rewards are typically not sparse. We also note that our extension of fully-amortized PB-BED to semi-amortized PB-BED is quite distinct to the typical generalization of offline RL approaches to hybrid RL approaches (Ball et al., 2023; Song et al., 2023; Zheng et al., 2022), as we are making refinements to the local policy during a single rollout using the same objective as original amortized policy. Our approach of refining a learned policy as new data becomes available also shares some similarities with Model Predictive Control (Qin and Badgwell, 2003).

# 6. Experiments

We empirically evaluate **Step-DAD** on a range of design problems, comparing its performance against **DAD** to determine the additional EIG achieved by the step policy $\pi^s$ over the fully amortized policy $\pi_0$. We further consider several other baselines for comparison. **Static** design learns a fixed set of designs prior to the experiment by optimising a PCE bound (Foster et al., 2020) that is equivalent to (6), but which is defined in terms of the designs rather than pol-

Table 1: **Source Location Finding.** Upper and lower bound estimates of total EIG. We report $\tau = 6$, rest in Table 11 in the Appendix. Errors show $\pm 1$s.e., computed over 16 (2048) histories for step methods (rest). DAD was trained for 50K steps, Step-DAD for 2.5K.

| Method | Lower bound ($\uparrow$) | Upper bound ($\downarrow$) |
|---|---|---|
| Random | $3.612 \pm 0.012$ | $3.613 \pm 0.012$ |
| Static | $3.945 \pm 0.026$ | $3.946 \pm 0.026$ |
| Step-Static ($\tau = 6$) | $3.974 \pm 0.008$ | $3.975 \pm 0.008$ |
| DAD | $7.040 \pm 0.012$ | $7.089 \pm 0.013$ |
| **Step-DAD** ($\tau = 6$) | $\mathbf{7.759 \pm 0.114}$ | $\mathbf{7.765 \pm 0.114}$ |

icy parameters (i.e. it learns a non-adaptive policy whose design choices are fixed). The **Step-Static** baseline is a two-stage approach that first trains a set of $\tau$ static designs by optimizing a PCE bound on $\mathcal{I}_{1 \to \tau}(\xi_1, \ldots, \xi_\tau)$, before approximating the posterior and training a new conditional set of static designs for the last $T - \tau$ steps by optimizing a PCE bound on $\mathcal{I}_{\tau+1 \to T}^{h_\tau}(\xi_{\tau+1}, \ldots, \xi_T)$. When possible, we include **Problem-Specific** baselines used for the relevant experiment before, instead considering a **Random** design strategy when not. In all cases, the number of contrastive samples used during training was $L = 1023$.

Our main metric for assessing the quality of various design strategies is the **total EIG**, $\mathcal{I}_{1 \to T}(\pi)$, as given in (2). For the baselines, we approximate it via a version of the sPCE lower bound (6) with $L = 10^5$ to ensure a tight bound, along with its upper bound counterpart—the sequential Nested Monte Carlo estimator (sNMC, Foster et al., 2021) (see Appendix B). For Step-DAD, we instead use a conservative lower bound estimate on its difference in performance compared to the original DAD network, before adding this to the corresponding DAD estimate (see Appendix C.3). This ensures any biases from using bounds instead of the true EIG lead to underestimation of the gains Step-DAD gives. Full details on experimental details are provided in Appendix C.

## 6.1. Source Location Finding

We first consider the source location finding experiment from Foster et al. (2021), which draws upon the acoustic energy attenuation model, detailed in Sheng and Hu (2005). The objective of the experiment is to infer the locations of some hidden sources using noisy measurements, $y$, of their combined signal intensity. Each source emits a signal that decreases in intensity according to the inverse-square law. A full description of the model is given in Appendix C.5.

We begin by learning a fully amortized DAD policy to perform $T = 10$ experiment steps to locate a single source. We chose a training budget of 50K gradient steps this policy, as we found that further training did not significantly improve performance with our chosen architecture.

**Single policy update** We systematically evaluate our Step-

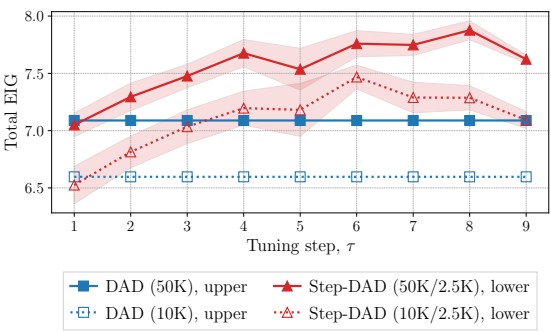

Figure 2: **Sensitivity to training budget** for location finding experiment. DAD policies are trained for 50K or 10K steps, Step-DAD policies are refined for 2.5K. Errors show $\pm 1$s.e.

DAD approach by exploring all possible fine-tuning steps, $\tau = 1, \ldots, 9$. We use importance sampling for posterior inference and fine-tune the policy for 2.5K steps. Results for $\tau = 6$ are presented in Table 1, whilst Table 11 in the Appendix shows performance for all values of $\tau$.

**Sensitivity to training budget** We investigate the overall resource efficiency of Step-DAD by comparing to DAD under two training budgets. The *full* budget is as before at 50K gradient steps, whilst the *reduced* budget is limited to 10K steps, both then followed by 2.5K finetuning steps for the Step-DAD networks. Figure 2 presents a conservative comparison, showing upper bound estimates for DAD and lower bound estimates for Step-DAD. The results reveal that Step-DAD consistently outperforms its respective DAD baseline for all $\tau > 1$ at both budget levels (the apparent slight drop for $\tau$ is likely due to the conservative estimation scheme used). Interestingly, Step-DAD with the reduced budget matches or exceeds the DAD with the full budget for all $\tau > 3$, thereby achieving better results with nearly 5 times fewer total training steps.

We note that the performance advantage of Step-DAD over DAD appears to be most pronounced when fine-tuning occurs just past the midpoint of the experiment, that is for $\tau = 6, 7$ or $8$. At this stage, our method can effectively leverage the accumulated data to refine the policy, while ensuring there are enough experiment steps remaining to benefit from the improved, customized policy.

**Multiple sources** We next consider a more complex setting of locating $2, 4$ and $6$ sources, which correspond to a 4-, 8- and 12-dimensional unknown parameter, respectively. Table 2 shows the results, indicating a consistent positive EIG difference for Step-DAD over DAD.

## 6.2. Robustness to prior perturbations

In BED, selecting an appropriate prior is critical as it both influences our final posterior, and the data we gather in the first place (Go and Isaac, 2022; Simchowitz et al., 2021).

Table 2: **Location finding multiple sources**. Reported for $\tau = 7$, Step-DAD (10K, 2.5K). Errors show $\pm 1$s.e.

| $\theta$ dim | EIG difference | DAD, total EIG (upper) |
|---|---|---|
| 4 | $0.701 \pm 0.023$ | $6.483 \pm 0.055$ |
| 8 | $0.426 \pm 0.014$ | $7.111 \pm 0.067$ |
| 12 | $0.423 \pm 0.012$ | $6.956 \pm 0.056$ |

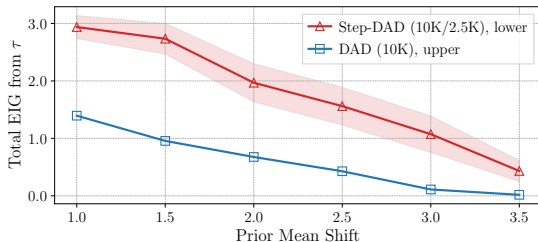

Figure 3: **Sensitivity to prior perturbations** in the location finding experiment. Total EIG for Step-DAD remains more robust compared to DAD, which drops to zero.

Fully amortized PB-BED approaches can be particularly prone to pathologies from imperfect prior choices, as the prior dictates the generated data the policy is trained on. Namely, if the prior predictive poorly matches the true data generating process (DGP), we may observe data at deployment that is highly distinct to anything seen in the policy training. To evaluate if Step-DAD can improve robustness to prior imperfections, we now consider a case where the prior, $p(\theta)$, used for the offline policy training leads to a DGP that is significantly different to the true DGP observed at deployment. To this end, we introduce a test-time prior $\tilde{p}(\theta)$, and evaluate the policy performance under the DGP $\tilde{p}(y_{1:T}|\xi_{1:T}) = \mathbb{E}_{\tilde{p}(\theta)}[\prod_{t=1}^{T} p(y_t|\theta, \xi_t, h_{t-1})]$, using the EIG under this *alternative model* as an evaluation metric

$$\mathcal{I}_{\tilde{p}(\theta)}(\pi) \coloneqq \mathbb{E}_{\tilde{p}(\theta)p(h_T|\theta,\pi)}\left[\log p(h_T \mid \theta, \pi) - \log \tilde{p}(h_T|\pi)\right],$$

where $\tilde{p}(h_T|\pi) = \mathbb{E}_{\tilde{p}(\theta)}[p(h_T|\theta, \pi)]$.

The results for the source location finding design problem are shown in Figure 3. It reveals that Step-DAD consistently outperforms the DAD baseline across all degrees of prior shift we consider, with the EIG for DAD decreasing to essentially zero with the increased prior shift, whilst Step-DAD is able to deliver positive information gains. This robustness is anticipated due to Step-DAD's ability to assimilate data gathered and adjust policies in light of new evidence.

## 6.3. Test-Time Compute Ablations

We now perform ablations to better understand how test-time performance is impacted by restrictions on computational budget. Firstly, for the location finding experiment we consider three different per-step budgets for the inference and then vary the amount of fine-tuning steps performed for updating the StepDAD network. As shown in Figure 4, total EIG generally improves with higher computational

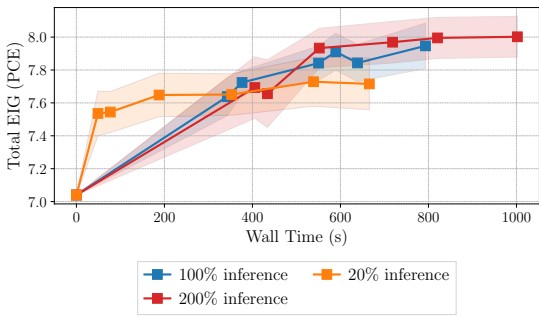

Figure 4: **EIG as a function of wall time** in the location finding experiment (T=10, $\tau = 6$), varying both the inference budget and the number of fine-tuning steps (between 250 and 10000 steps). Here 100% inference budget corresponds to taking 20000 importance samples.

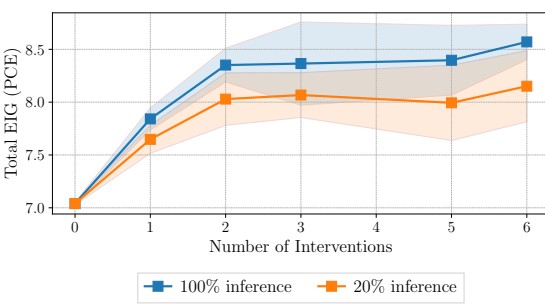

Figure 5: **Sensitivity to number of interventions** for location finding experiment (T=10) for different inference budgets (100% = 20000 importance samples). EIG increases with more interventions, then plateaus. Each update corresponds to posterior inference + 2.5k fine-tuning steps.

budgets, with diminishing returns for large budgets. The performance for the 100% and 200% inference budgets are quite similar, indicating our inference has been successful, but a performance drop is seen when only using 20% of our previous inference budget. Importantly though, significant gains relative to DAD (corresponding to $0s$ wall time in the plot) are still achieved with small budgets corresponding to around a minute of wall time.

Secondly, we evaluate how performance evolves as a function of the number of interventions. Each intervention consists of updating the posterior and applying 2.5k fine-tuning steps. Figure 5 shows that Total EIG consistently increases with more interventions across all inference budgets, before eventually plateauing. We note that even for a small number of interventions, there is increased performance over DAD (corresponding to 0 interventions in the plot), demonstrating the utility of Step-DAD even in settings of constrained test-time compute budget.

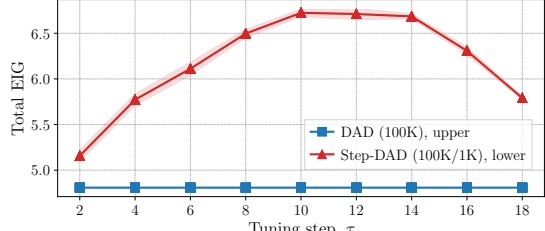

Figure 6: **Hyperbolic temporal discounting.** EIG improvement of Step-DAD over DAD after fine-tuning the policy at step $\tau$. The fully amortized DAD policy is trained for 100K steps, step-policy is refined for 1K steps.

Table 3: **Hyperbolic temporal discounting.** Estimates of total EIG, $\mathcal{I}_{1\to20}(\pi)$. Errors indicate $\pm$ 1 s.e., ran over 16 (2048) histories for the step methods (rest). Baselines as reported in Foster et al. (2021), except DAD and Random.

| Method | Lower bound ($\uparrow$) | Upper bound ($\downarrow$) |
|---|---|---|
| Random | $2.249 \pm 0.010$ | $2.249 \pm 0.010$ |
| Kirby (2009) | $1.861 \pm 0.008$ | $1.864 \pm 0.009$ |
| Static | $2.518 \pm 0.007$ | $2.524 \pm 0.007$ |
| Frye et al. (2016) | $3.500 \pm 0.029$ | $3.513 \pm 0.029$ |
| Greedy (BADapted) | $4.454 \pm 0.016$ | $4.536 \pm 0.018$ |
| DAD | $4.778 \pm 0.013$ | $4.808 \pm 0.014$ |
| **Step-DAD** ($\tau$=10) | $\mathbf{6.711 \pm 0.040}$ | $\mathbf{6.721 \pm 0.040}$ |

### 6.4. Hyperbolic Temporal Discounting

Temporal discounting describes the tendency for individuals to prefer smaller immediate rewards over larger delayed ones. This phenomenon is a key concept in psychology and economics and has been used to study important social and individual behaviors (Critchfield and Kollins, 2001), including dietary choices (Bickel et al., 2021), exercise habits (Tate et al., 2015), patterns of substance abuse and addictive behaviours (Story et al., 2014). An individual's time delay preference is typically measured by asking them a series of questions, such as "Would you prefer \$$R$ now or \$100 in $D$ days time?" Here the tuple $\xi = (R, D)$ defines our experimental design, and the experiment outcome $y$ is the participant's decision to either *accept* or *reject* the delay.

**Single update** Using the hyperbolic discounting model introduced in Mazur (1987) and as implemented by Vincent (2016), we train a DAD policy for 100K gradient steps, aimed at designing $T = 20$ experiments. We select a grid of tuning steps $\tau$ in the range from 2 to 18 in increments of 2. For posterior inference, we use simple importance sampling to draw samples from the posterior $p(\theta \,|\, h_\tau)$ and 1% of the original training budget (i.e. 1K gradient steps).

Figure 6 reports the results and illustrates that Step-DAD yields an improvement in total EIG for *all* choices of $\tau$ when compared to the baseline DAD policy. The largest increase occurs around, and shortly after, the middle of the experiment, aligning with our previous intuition: here sufficient

data has been accumulated to inform a meaningful posterior update, whilst sufficient number of experiments remain to effectively deploy the refined policy. Table 3 demonstrates the superiority of Step-DAD over conventional baselines, including those derived from psychology research (Frye et al., 2016; Kirby, 2009; Vincent and Rainforth, 2017) and traditional BED approaches such as the BADapted (Vincent and Rainforth, 2017) approach which was specifically designed for this problem. It also outperforms the static BED strategy, highlighting the effectiveness of adaptive design strategies in extracting more valuable information from experiments.

**Multiple updates and design extrapolation** We extend the deployment of DAD and Step-DAD policies for this problem to $T = 40$ experiment steps, doubling the scope at which they were originally trained, i.e. without retraining the DAD network. Step-DAD is fine-tuned at two steps, $\tau$ and $2\tau$, with $\tau \in \{5, 6, 7, 8\}$. As Table 4 shows, Step-DAD demonstrates significantly improved capacity to extract information in later stages, beyond its initial training. This highlights the robustness and flexibility of our method in extending experimental horizons compared to DAD.

### 6.5. Constant Elasticity of Substitution (CES)

We conclude our evaluation with the Constant Elasticity of Substitution (CES) model, a framework from behavioral economics to analyse the relative utility of two baskets of goods (Arrow et al., 1961). This model emulates how economic actors specify their relative preference $y$ between these baskets on a sliding scale. We follow the experimental setup of Foster et al. (2019), with full details given in Appendix C.7. The CES model faces challenges due to $y$ being sampled from a censored normal distribution, concentrating probability at observation boundaries and creating local maxima (Blau et al., 2022; Foster et al., 2019).

Table 5 shows that Step-DAD outperforms all baselines. Step-Static also achieves on-par results with Greedy (vPCE), highlighting the benefits of semi-amortized design strategies in this model. We note the performance of DAD is worse than Step-Static. This can be attributed to the discontinuities in the censored likelihood (Eq. (29) in the Appendix) that complicate the training of the policy, often resulting in

Table 5: **Constant Elasticity of Substitution.** Estimates on total EIG, $\mathcal{I}_{1 \to 10}(\pi)$. DAD and Static trained for 50K steps; step variants finetuned for 10K steps. Errors denote $\pm$ 1 s.e.

| Method | Lower bound ($\uparrow$) | Upper bound ($\downarrow$) |
|---|---|---|
| Random | $2.487 \pm 0.007$ | $2.487 \pm 0.007$ |
| Greedy (vPCE) | $13.333 \pm 0.975$ | $13.343 \pm 0.975$ |
| Static | $9.279 \pm 0.020$ | $11.183 \pm 0.0453$ |
| Step-Static ($\tau = 5$) | $13.010 \pm 0.185$ | $13.682 \pm 0.189$ |
| DAD | $10.181 \pm 0.021$ | $11.478 \pm 0.042$ |
| **Step-DAD ($\tau = 5$)** | **$13.879 \pm 0.352$** | **$14.623 \pm 0.363$** |

convergence to suboptimal local maxima.

## 7. Conclusions

In this work, we introduced the idea of a semi-amortized approach to PB-BED that enhances the flexibility, robustness and effectiveness of fully amortized design policies. Our method, Stepwise Deep Adaptive Design (Step-DAD), dynamically updates its step policy in response to new data through a systematic 'infer-refine' procedure that refines the design policy for the remaining experiments in light of the experimental data gathered so far. This iterative refinement enables the step policy to evolve as the experiment progresses, ensuring more robust and tailored design decisions, as demonstrated in our empirical evaluation. Step-DAD thus improves our ability to conduct more efficient, informed, and robust experiments, opening new avenues for exploration in various scientific domains.

## Impact Statement

This paper presents work whose goal is to advance the field of Machine Learning. There are many potential societal consequences of our work, none which we feel must be specifically highlighted here.

## Acknowledgements

MH is supported by funding provided by Novo Nordisk and by the EPSRC Centre for Doctoral Training in Modern Statistics and Statistical Machine Learning. TR is supported by the UK EPSRC grant EP/Y037200/1.

Table 4: **Hyperbolic temporal discounting: extrapolating designs.** Comparison of EIG upper bound for DAD and lower bound for Step-DAD across tuning steps $\tau$ and $T = 40$. Errors indicate $\pm 1$s.e., computed over 16 histories.

| | EIG from $\tau$ ($\uparrow$) | | EIG from $2\tau$ ($\uparrow$) | |
|---|---|---|---|---|
| $\tau$ | DAD | Step-DAD | DAD | Step-DAD |
| 5 | $3.9 \pm 0.24$ | **$4.8 \pm 0.14$** | $2.1 \pm 0.36$ | **$4.6 \pm 0.18$** |
| 6 | $3.4 \pm 0.30$ | **$5.1 \pm 0.07$** | $1.7 \pm 0.30$ | **$4.7 \pm 0.12$** |
| 7 | $3.0 \pm 0.35$ | **$4.4 \pm 0.30$** | $1.3 \pm 0.23$ | **$4.4 \pm 0.11$** |
| 8 | $2.6 \pm 0.36$ | **$4.7 \pm 0.13$** | $1.0 \pm 0.19$ | **$4.2 \pm 0.13$** |

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

# A. Algorithm

---

**Algorithm 1** Overview of Step-DAD

---

**Input:** Generative model $p(\theta)p(y \mid \theta, \xi)$, experimental budget $T$, refinement schedule $\mathcal{T} = \{\tau_0, \tau_1, \ldots, \tau_{K+1}\}$, with $\tau_0 = 0, \tau_{K+1} = T$, training budgets $\{N_{\tau_k}\}_{k=1:K}$

**Output:** Dataset $h_T = \{(\xi_t, y_t)\}_{t=1:T}$

OFFLINE STAGE: BEFORE THE LIVE EXPERIMENT
$\quad \triangleright$ Set $h_0 = \varnothing$.
$\quad$ **while** Computational budget does not exceed $N_0$ **do**
$\quad\quad \triangleright$ Train fully-amortized $\pi_0$ as in Foster et al. (2021)
$\quad$ **end**
ONLINE STAGE: DURING THE LIVE EXPERIMENT
$\quad$ **for** $k = 1, \ldots, K+1$ **do**
$\quad\quad$ **for** $\tau_{k-1} < t \leq \tau_k$ **do**
$\quad\quad\quad \triangleright$ Compute design $\xi_t = \pi_{\tau_{k-1}}(h_{t-1})$
$\quad\quad\quad \triangleright$ Run experiment $\xi_t$, observe an outcome $y_t$
$\quad\quad\quad \triangleright$ Update the dataset $h_t = h_{t-1} \cup (\xi_t, y_t)$
$\quad\quad$ **end**
$\quad\quad$ **If** $k = K+1$ **then return** $h_T$ **end**
$\quad\quad$ **while** Computational budget does not exceed $N_k$ **do**
$\quad\quad\quad \triangleright$ Fit a posterior $p(\theta \mid h_{\tau_k})$
$\quad\quad\quad \triangleright$ Fine-tune policy $\pi_{\tau_k}$ by optimizing (6)
$\quad\quad$ **end**
$\quad$ **end**

---

# B. Further EIG bounds

The sequential Nested Monte Carlo (sNMC) (Foster et al., 2021) upper bound is given by

$$\mathcal{U}_{\tau_k+1 \to T}^{h_{\tau_k}}(\pi) := \mathbb{E}\left[\log \frac{p(h_{\tau_k+1:T} \mid h_{\tau_k}\theta_0, \pi)}{\frac{1}{L}\sum_{\ell=1}^{L} p(h_{\tau_k+1:T} \mid \theta_\ell, \pi)}\right], \tag{7}$$

which we use to evaluate different design strategies.

For implicit models we can utilize the InfoNCE bound (van den Oord et al., 2018), which is given by

$$\mathcal{L}_{\text{InfoNCE}}(\pi, U) := \mathbb{E}_{p(\theta_0)p(h_T \mid \theta_0, \pi)}\mathbb{E}_{p(\theta_{1:L})}\left[\log \frac{\exp(U(h_T, \theta_0))}{\frac{1}{L+1}\sum_{i=0}^{L}\exp(U(h_T, \theta_i))}\right], \tag{8}$$

or the NWJ bound (Nguyen et al., 2010), given by:

$$\mathcal{L}_{\text{NWJ}}(\pi, U) := \mathbb{E}_{p(\theta)p(h_T \mid \theta, \pi)}\left[U(h_T, \theta) - e^{-1}\mathbb{E}_{p(\theta)p(h_T \mid \pi)}\left[\exp(U(h_T, \theta))\right]\right], \tag{9}$$

where in both bounds $U$ is a learnt critic function, $U : h_T \times \theta \mapsto \mathbb{R}$.

# C. Experiment details

## C.1. Computational resources

The experiments were conducted using Python and open-source tools. PyTorch (Paszke et al., 2019) and Pyro (Bingham et al., 2018) were employed to implement all estimators and models. Additionally, MlFlow (Zaharia et al., 2018) was utilized for experiment tracking and management. Experiments were performed on two separate GPU servers, one with 4xGeForce RTX 3090 cards and 40 cpu cores; the other one with 10xA40 and 52 cpu cores. Every experiment was ran on a single GPU.

## C.2. Policy architecture

In the same vein as Foster et al. (2021), we leverage the permutation invariance of EIG in our BED problem settings to allow for more efficient network training and *weight sharing*. That is to say we represent histories of varying lengths by a single fixed dimensional representations $R(h_t)$ parameterized by an *encoder network* $E_{\phi_1}$ (inheriting notation from Foster et al. (2021)),

$$R(h_t) := \sum_{k=1}^{t} E_{\phi_1}(\xi_k, y_k). \tag{10}$$

Our policy then becomes: $\pi_\phi(h_t) = F_{\phi_2}(R(h_t))$ where $F_{\phi_2}$ is a decoder network. The specific architectures for the encoder and decoder networks are given in Sections C.5 and C.6.

## C.3. Evaluation details

When evaluating fully amortized policies, we employ the sPCE (6) lower bound and sNMC (7) upper bound using a large number of contrastive samples, $L = 100K$, drawn from the prior to approximate the inner expectation. The outer expectation is approximated using $N = 2048$ draws from the model $p(\theta)p(h_T \mid \theta, \pi)$. To approximate the total EIG quantity for Step-DAD efficiently, we use

$$\Delta \mathcal{I}(\pi^s, \pi_0) := \mathcal{I}_{1 \to T}(\pi^s) - \mathcal{I}_{1 \to T}(\pi_0) \tag{11}$$

$$= \mathbb{E}_{p(h_\tau \mid \pi_0)}[\mathcal{I}_{\tau+1 \to T}^{h_\tau}(\pi^s) - \mathcal{I}_{\tau+1 \to T}^{h_\tau}(\pi_0)] \tag{12}$$

$$\geq \mathbb{E}_{p(h_\tau \mid \pi_0)} \left[ \mathcal{L}_{\tau+1 \to T}^{h_\tau}(\pi^s) - \mathcal{U}_{\tau+1 \to T}^{h_\tau}(\pi_0) \right], \tag{13}$$

and add that difference to our corresponding lower/upper bound estimates of $\mathcal{I}_{1 \to \tau}(\pi_0)$ (which can be directly estimated using the sPCE and sNMC bounds respectively). Here we are using a combination of an upper and lower bound is done to ensure any reported gains from StepDAD are conservatively underestimated, as well as reducing the variance in our estimates.

Note, as found in Blau et al. (2022), the lower and upper bounds are less tight compared to other experiments. In this experiment, total EIG was simply calculated with the higher variance alternative as follows: $EIG_{1 \to \tau}(\pi_0) + \mathbb{E}_{p(h_\tau \mid \pi_0)}[EIG_{\tau \to T}(\pi^s)]$.

## C.4. Baselines

**Static** The Static (fixed) baseline pre-selects a fixed $\xi_1, ..., \xi_T$ ahead of the experiment before any observations. As in all cases, designs are optimized to maximise the EIG. This non-adaptive approach used PCE bound to optimize the design set $\xi_1, ..., \xi_T$ and can be thought of treating the entire sequence of experiments as a single experiment (Foster et al., 2021).

**Step-Static** Step-Static computes a set of designs for $\xi_1, ..., \xi_\tau$ before a posterior update and subsequent computation of designs $\xi_\tau, ..., \xi_T$. Each set of designs are selected following the static methodology outlined above.

**Random** As the name implies, this baseline selects a random sample of designs $\xi_1, ..., \xi_T$. Thus the most non-informed naive approach.

## C.5. Location Finding

The objective of the experiment is to ascertain the location, $\theta$, of $K$ sources. $K$ is presumed to be predetermined. The intensity at each selected design choice, $\xi$, represents a noisy observation $\log y \mid \theta, \xi$ centered around the logarithm of the underlying model, $\mu(\theta, \xi)$

$$\mu(\theta, \xi) = b + \sum_{k=1}^{K} \frac{\alpha_k}{(m + ||\theta_k - \xi||)^2}. \tag{14}$$

In the given context, $\alpha_k$ may be either predetermined constants or random variables, $b > 0$ represents a fixed background signal, and $m$ is a constant representing maximum signal

$$\log[y \mid \theta, \xi] \sim \mathcal{N}(\log \mu(\theta, \xi), \sigma^2). \tag{15}$$

We assumed a normal standard prior at training: $\theta_k \overset{\text{i.i.d.}}{\sim} \mathcal{N}(0_d, I_d)$.

### C.5.1. TRAINING DETAILS

Tables 6 and 7 outline the architecture of the DAD policy network. The model hyperparameters used are outlined in Tables 8, 9 and 10.

Table 6: **Source location finding.** Encoder network $E_{\phi_1}$, architecture as in Foster et al. (2021)

| Layer | Overview | Dimension | Activation |
|---|---|---|---|
| Design-outcome | $\xi, y$ | 3 | - |
| H1 | Fully connected | 64 | RELU |
| H2 | Fully connected | 256 | RELU |
| Output | Fully Connected | 16 | - |

Table 7: **Source location finding.** Decoder network $F_{\phi_2}$, architecture as in Foster et al. (2021)

| Layer | Overview | Dimension | Activation |
|---|---|---|---|
| Input | $E(h_t)$ | 16 | - |
| H1 | Fully connected | 128 | RELU |
| H1 | Fully connected | 16 | RELU |
| Output | $\xi$ | 2 | - |

Table 8: **Source location finding.** Parameter Values

| Parameter | Value |
|---|---|
| $\alpha_k$ | 1 for all $k$ |
| Max signal, $m$ | $10^{-4}$ |
| Base signal, $b$ | $10^{-1}$ |
| Observation noise scale, $\sigma$ | 0.5 |

Table 9: **Source location finding.** Parameters for training pre-training DAD/Step-DAD

| Parameter | Value |
|---|---|
| Batch size | 1024 |
| Number of negative samples | 1023 |
| Number of gradient steps (default) | 50K |
| Learning rate (LR) | 0.0001 |

Table 10: **Source location finding.** Parameters for Step-DAD finetuning

| Parameter | Value |
|---|---|
| Number of theta rollouts | 16 |
| Number of posterior samples | 20K |
| Finetuning learning rate (LR) | 0.0001 |

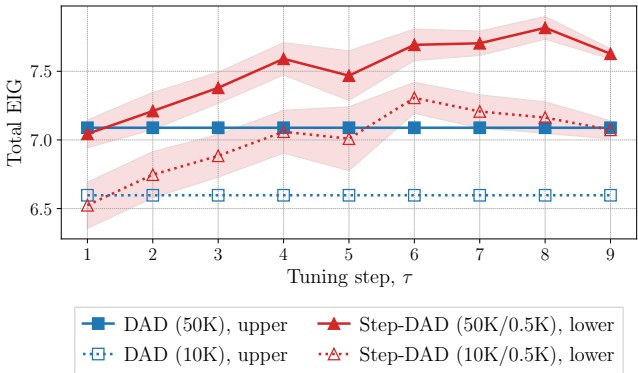

Figure 7: **Sensitivity to training budget** for location finding experiment. DAD policies are trained for 50K or 10K steps, Step-DAD policies are refined for 0.5K steps. Step-DAD outperforms its respective DAD baseline for all $\tau$. Errors show $\pm 1$s.e.

Table 11: **Source location finding.** Total EIG for Step-DAD for various tuning steps $\tau$. Base DAD network was pretrained for 50k steps before subsequent 2.5k finetuning steps at $\tau$.

| $\tau$ | Lower bound ($\uparrow$) | Upper bound ($\downarrow$) |
|---|---|---|
| **1 (Worst)** | **7.050 ($\pm$ 0.103)** | **7.065 ($\pm$ 0.103)** |
| 2 | 7.295 ($\pm$ 0.119) | 7.305 ($\pm$ 0.120) |
| 3 | 7.476 ($\pm$ 0.102) | 7.482 ($\pm$ 0.103) |
| 4 | 7.676 ($\pm$ 0.118) | 7.680 ($\pm$ 0.118) |
| 5 | 7.536 ($\pm$ 0.182) | 7.540 ($\pm$ 0.182) |
| 6 | 7.759 ($\pm$ 0.114) | 7.765 $\pm$ 0.114 |
| 7 | 7.748 ($\pm$ 0.091) | 7.757 ($\pm$ 0.091) |
| **8 (Best)** | **7.877 ($\pm$ 0.082)** | **7.892 ($\pm$ 0.082)** |
| 9 | 7.623 ($\pm$ 0.035) | 7.652 ($\pm$ 0.035) |
| DAD | 7.040 ($\pm$ 0.012) | 7.089 ($\pm$ 0.013) |

### C.5.2. OPTIMAL $\tau$

The optimal value for EIG occurs at a range around $\tau \in [6, 7, 8]$ (Table 11). Figure 7 demonstrates a further ablation from Section 6.1 where we now finetune the Step-DAD policies for 0.5K steps instead of 2.5K. We observe the same behavior for this reduced finetuning as before. Additionally, we note Step-DAD under the *reduced* 10K pretraining budget still outperforms DAD under the *full* 50K budget for $\tau = 6$.

### C.5.3. MULTIPLE SOURCES

As a further ablation, we test the robustness of a semi-amortized approach to the more complex task of location finding with multiple sources of signal. We find a positive EIG difference in all cases, once again demonstrating the benefits of using the semi-amortized Step-DAD network compared to the baseline fully amortized DAD network. Increasing the number of sources leads to a reduction in the EIG difference. However, this is expected given the increasing complexity of the task compared to the fixed number of steps post $\tau$ to adjust the decision making policy in the semi amortized setting. All experiments were run with $\tau = 7$. Refer to Table 2 in main paper for results.

### C.6. Hyperbolic Temporal Discounting Model

Building on Foster et al. (2021), Mazur (1987) and Vincent (2016), we consider a hyperbolic temporal discounting model. A participant's behaviour is characterized by the latent variables $\theta = (k, \alpha)$ with prior distributions as follows:

$$\log k \sim \mathcal{N}(-4.25, 1.5) \quad \alpha \sim \text{HalfNormal}(0, 2). \tag{16}$$

HalfNormal distribution denotes a Normal distribution truncated at 0. For given $k, \alpha$, the value of the two propositions "£R today" and "£100 in D days" with design $\xi = (R, D)$ are given by:

$$V_0 = R, \qquad V_1 = \frac{100}{1 + kD}. \tag{17}$$

Participants select $V_1$ in place of $V_0$ with probability modelled as:

$$p(y = 1 | k, \alpha, R, D) = \epsilon + (1 - 2\epsilon)\Phi\left(\frac{V_1 - V_0}{\alpha}\right). \tag{18}$$

We fix $\epsilon = 0.01$ and $\phi$ is the c.d.f of the standard Normal Distribution

$$\Phi(z) = \int_{-\infty}^{z} \frac{1}{\sqrt{2\pi}} \exp{-\frac{1}{2}z^2}. \tag{19}$$

As in Foster et al. (2021), the design parameters $R, D$ have the constraints $D > 0$ and $0 < R < 100$. $R, D$ are represented in an unconstrained space $\xi_d, \xi_r$ and transformed using the below maps

$$D = \exp(\xi_d) \qquad R = 100 \cdot \text{sigmoid}(\xi_r). \tag{20}$$

Tables 12 and 13 outline the architecture of the DAD policy network. Tables 20 and 15 give the hyper-parameters for training the DAD/Step-DAD policies for the hyperbolic temporal discounting model.

Table 12: **Hyperbolic Temporal Discounting model.** DAD encoder network.

| Layer | Overview | Dimension | Activation |
|---|---|---|---|
| Design input | $\xi_d, \xi_r$ | 2 | - |
| H1 | Fully connected | 256 | Softplus |
| H2 | Fully connected | 256 | Softplus |
| H3 | Fully connected | 16 | - |
| H3' | Fully connected | 16 | - |
| Output | $y \odot H3 + (1 - y) \odot H3'$ | 16 | - |

Table 13: **Hyperbolic Temporal Discounting model.** DAD decoder (emission) network

| Layer | Overview | Dimension | Activation |
|---|---|---|---|
| Input | $R(h_t)$ | 16 | - |
| H1 | Fully connected | 256 | Softplus |
| H2 | Fully connected | 256 | Softplus |
| Output | $\xi_d, \xi_r$ | 2 | - |

Table 14: **Hyperbolic Temporal Discounting model.** Parameters for training of the DAD network.

| Parameter | Value |
|---|---|
| Batch size | 1024 |
| Number of negative samples | 1023 |
| Number of gradient steps (default) | 100K |
| Learning rate (LR) | $5 \times 10^{-5}$ |
| Annealing frequency | 1K |
| Annealing factor | 0.95 |

Table 15: **Hyperbolic Temporal Discounting model.** Parameters for Step-DAD policy finetuning.

| Parameter | Value |
|---|---|
| Number of theta rollouts | 16 |
| Number of posterior samples | 20K |
| Finetuning learning rate (LR) | $5 \times 10^{-5}$ |

## C.7. Constant Elasticity of Substitution (CES)

This experiment builds upon a behavioral economic model in which participants compare the differing utility, $U(x)$, of two presented baskets of goods $x$ (Arrow et al., 1961). In this experiment $x \in [0, 100]^3$ represents non-negative quantities of three goods which together form the basket for evaluation.

The agent compares the two baskets $x$ and $x'$ by evaluating their individual utility $U(x)$ and subsequently indicating their preference using a sliding scale ranging from 0 to 1 following the probabilistic model defined below. The latent variables governing this framework would in practice vary across individuals; representing their unique preferences. Thus the experimental design objective is to infer each of these latent parameters and therefore understand the individual preference model of the decision maker in question. The priors are defined as follows:

$$\rho \sim \text{Beta}(1, 1) \tag{21}$$
$$\alpha \sim \text{Dirichlet}([1, 1, 1]) \tag{22}$$
$$\log u \sim \mathcal{N}(1, 3) \tag{23}$$
$$\tag{24}$$

The probabilistic model is expressed as:

$$U(x) = \left( \sum_i x_i^\rho \alpha_i \right)^{1/\rho} \tag{25}$$
$$\mu_\eta = (U(x) - U(x'))u \tag{26}$$
$$\sigma_\eta = (1 + \|x - x'\|)\tau \cdot u \tag{27}$$
$$\eta \sim \mathcal{N}(\mu_\eta, \sigma_\eta^2) \tag{28}$$
$$y = \text{clip}(\text{sigmoid}(\eta), \epsilon, 1 - \epsilon) \tag{29}$$

The values of the hyper-parameters used in the model are detailed in Table 16.

Table 16: **CES model.** Hyper-parameter values.

| Parameter | Value |
|---|---|
| $\tau$ | 0.005 |
| $\epsilon$ | $2^{-22}$ |

Table 17: **CES model.** DAD embedding layers. Outcomes and Designs are embedded and then concatenated before being passed into the encoder.

| Layer | Overview | Dimension | Activation |
|---|---|---|---|
| Input | $\xi_d$ or $y$ | 6 or 1 | - |
| Output | Fully connected (Layer Norm) | 32 | ReLU |

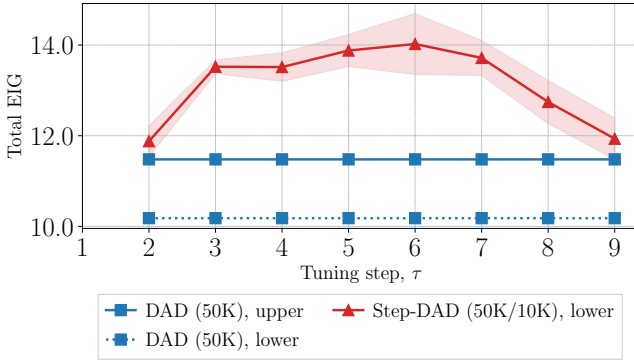

Figure 8: **Constant Elasticity of Substitution.** Improvement in EIG by Step-DAD over DAD after fine-tuning. DAD is trained for 50K steps, with Step-DAD undergoing an additional 10K finetuning steps. Step-DAD (lower) outperforms the DAD baseline. DAD was initialized with designs from the Static methodology to overcome challenges introduced by the accumulated probability mass at the boundaries following censoring.

Table 18: **CES model.** DAD encoder network.

| Layer | Overview | Dimension | Activation |
|---|---|---|---|
| Input | Embedding | 64 | - |
| H1 | Fully connected (Layer Norm) | 128 | ReLU |
| Output | Fully connected (Layer Norm) | 32 | ReLU |

Table 19: **CES model.** DAD decoder (emission) network

| Layer | Overview | Dimension | Activation |
|---|---|---|---|
| Input | Embedding | 32 | - |
| H1 | Fully connected | 128 | ReLU |
| Output | Fully connected | 6 | - |

Table 20: **CES model.** Parameters for training of the DAD network.

| Parameter | Value |
|---|---|
| Batch size | 100 |
| Number of negative samples | 1024 |
| Number of gradient steps (default) | 50K |
| Learning rate (LR) | $1 \times 10^{-4}$ - $1 \times 10^{-5}$ |
| Annealing frequency | 1K |

Table 21: **CES model.** Parameters for Step-DAD policy finetuning.

| Parameter | Value |
|---|---|
| Number of theta rollouts | 16 |
| Number of posterior samples | 20K |
| Finetuning learning rate (LR) | $1 \times 10^{-5}$ |

