# OpenReview forum: "Step-DAD: Semi-Amortized Policy-Based Bayesian Experimental Design"
_ICML.cc/2025/Conference — ICML 2025 poster_

### Official Review · Reviewer_EbFb · 2025-03-08

**Overall Recommendation:** 3

**Summary:**

This paper proposes Step-DAD, a semi-amortized approach to Bayesian experimental design (BED) that extends the existing Deep Adaptive Design (DAD) framework. While DAD pre-trains a fixed policy network before experimentation, Step-DAD allows for test-time adaptation of this policy during deployment, periodically refining it based on accumulated experimental data.

The authors evaluate Step-DAD on three experimental design problems: source location finding, hyperbolic temporal discounting, and constant elasticity of substitution. Results consistently show that Step-DAD outperforms DAD and other BED baselines, including when dealing with prior misspecification and extended experimental horizons.

**Claims And Evidence:**

The paper's main claim that Step-DAD outperforms existing BED methods is well-supported by empirical evidence across multiple experimental settings, which are commonly used in BED literature. The authors present thorough comparisons with appropriate baselines using consistent evaluation metrics (EIG bounds).

**Essential References Not Discussed:**

It would be nice to also mention some other latest BED work in the related work section, where some of these approaches can benefit from the proposed semi-amortized framework:

[1] Iollo, Jacopo, et al. "Bayesian Experimental Design Via Contrastive Diffusions." ICLR.

[2] Iollo, Jacopo, et al. "PASOA-PArticle baSed Bayesian optimal adaptive design." ICML.

[3] Huang, Daolang, et al. "Amortized Bayesian Experimental Design for Decision-Making." Neurips.

[4] Iqbal, Sahel, et al. "Nesting Particle Filters for Experimental Design in Dynamical Systems." ICML.

[5] Iqbal, Sahel, et al. "Recursive Nested Filtering for Efficient Amortized Bayesian Experimental Design." Neurips BDU workshop.

**Experimental Designs Or Analyses:**

The experimental designs are generally sound. However, it would be nice to see a comparison with RL-based BED approaches such as the work by Blau et al. (2022). Given that the semi-amortized framework is architecture-agnostic, it would be valuable to see how Step-DAD principles could be applied to RL-BED methods. This would provide a more comprehensive understanding of the benefits of semi-amortization across different policy learning paradigms.

**Methods And Evaluation Criteria:**

The proposed methods are appropriate for the BED problem. Using EIG bounds as the evaluation metric aligns with the standard objective in BED literature.

The problems selected in the paper are appropriate, which are commonly used benchmarks in BED literature. The inclusion of additional evaluation scenarios such as prior misspecification strengthens the assessment of the method's practical utility.

However, one minor concern is that all evaluations are on synthetic problems rather than real-world datasets, it would be nice to see one such case which can support the need of doing test-time training.

**Other Comments Or Suggestions:**

The paper is very well written, I can't spot any typos or apparent mistakes.

**Other Strengths And Weaknesses:**

Strengths:
* The semi-amortized framework is a natural extension of existing BED approaches.
* The empirical improvements are significant and consistent across problems.
* The paper is well-written and easy to understand.

Weaknesses:
* The contribution is somewhat incremental, building directly on DAD without fundamental architectural innovations.
* While the paper demonstrates robustness to prior misspecification, it lacks exploration of other types of model misspecification. Given that the framework can be used to address model misspecification, investigating additional forms of misspecification (e.g., likelihood function misspecification) would strengthen the paper's claims about robustness.
* The paper lacks a systematic analysis for balancing additional computational cost against EIG gains. A metric or guideline for determining the optimal update schedule (number and timing of updates) would provide practical value for deployment.

**Questions For Authors:**

* For real-world applications where online policy updates might be computationally constrained, do you have recommendations for determining the optimal update schedule (when and how often to update)?
* The paper shows Step-DAD is more robust to prior misspecification. Have you investigated how it performs when other aspects of the model are misspecified (e.g., likelihood functions)?

**Relation To Broader Scientific Literature:**

The paper properly situates itself within the BED literature, acknowledging connections to:

* Traditional adaptive BED frameworks
* Policy-based BED approaches
* Reinforcement learning (RL) literature, particularly offline model-based RL

**Theoretical Claims:**

I've checked the theoretical claims, particularly Proposition 3.1 regarding the decomposition of total EIG. The proof appears sound, correctly leveraging the factorization of joint likelihoods and probabilities to establish the independence of policy optimality for later steps given the history at the intermediate point.

---

> ### Author Rebuttal · Authors · 2025-03-31
>
> Thank you kindly for your helpful review.
>
> >It would be nice to see a comparison with RL-based BED approaches such as the work by Blau et al. (2022). Given that the semi-amortized framework is architecture-agnostic, it would be valuable to see how Step-DAD principles could be applied to RL-BED methods.
>
> We agree that this could be a nice addition if time and space allow. As you note, the StepDAD approach can be directly applied within RL-BED policy training frameworks, so these simply provide an alternative strategy for the policy refinement we are proposing, rather than a directly competing approach. The main reason we focused on direct policy training strategies in our experiments was that they generally allow the policy to be updated far more cheaply than using an RL-based approach, thus making them more suitable for deployment-time policy refinements where time is more critical than the original policy training. We will add further discussion on this to the paper.
>
> >It would be nice to also mention some other latest BED work in the related work section, where some of these approaches can benefit from the proposed semi-amortized framework:
>
> Thank you for this excellent suggestion. We will update the related work section to include discussion of the additional papers you mention, several of which could benefit from or be extended by the proposed semi-amortized framework.
>
> > The paper lacks a systematic analysis for balancing additional computational cost against EIG gains.
>
> Thank you for highlighting this. We agree that this analysis would benefit the paper and have added new results on the trade-off between cost and performance. Specifically, we have done two new ablations for the location finding experiment. The first is to vary the amount of compute spent on inference and fine-tuning. The results, which can be found here: https://tinyurl.com/EIG-wall-time (anonymous), show that meaningful gains over DAD can be achieved with only a couple of minutes of computation. The benefits improve as the fine-tuning budget is increased, before plateauing.
>
> The second new ablation is to increase the number of times that the policy is refined through the experiment. The results can be found here: https://tinyurl.com/eig-interventions (anonymous). They show that there is an initial benefit to increasing the number of intervention steps before a plateauing once again.
>
> >For real-world applications where online policy updates might be computationally constrained, do you have recommendations for determining the optimal update schedule (when and how often to update)?
>
> While the optimal update schedule will vary between applications, our findings do provide some helpful guidance:
> - Our new results above show that there are often diminishing returns from conducting many policy refinements, so it will not usually be necessary to refine the policy at every step, in fact there is a general plateau beyond two interventions for the T=10 budget case.
>
> - Figures 2 and 4 suggest that refining the policy in the region of halfway to three quarters of the way through the experiment may have the biggest impact when only doing a single refinement.
>
> - Even small amounts of policy refinement (on the order of a couple of minutes) can be beneficial.
>
> We also note that in many real applications, the update schedule may be directly dictated by the computate constraints, with little flexibility in the update schedule.  For example, if each experiment itself takes, ~10 minutes to run, this gives us a clear per-iteration budget that we can use while we wait for the next experiment to complete. We will add further discussion on these practical considerations.
>
> >The paper shows Step-DAD is more robust to prior misspecification. Have you investigated how it performs when other aspects of the model are misspecified (e.g., likelihood functions)?
>
> Thank you for the great suggestions. We have not explicitly investigated robustness to likelihood misspecification. Like other amortized BED methods, Step-DAD remains sensitive in such cases, and understanding this susceptibility is an important direction for future work in BED literature. We believe that Step-DAD in its current form is less likely to help guard against likelihood misspecification than prior misspecification as the same likelihood is still used in the policy refinement, whereas the prior is replaced by the intermediary posterior (which may have corrected some of the issues of the original prior, such as if it is not sufficiently informative).
>
> Interestingly though, Step-DAD does open up avenues for future work in this direction.  For example, one could consider doing model checking before policy refinement, then training under a new likelihood if the data collected indicates the original one should be rejected. We feel it is beyond the scope of the current paper to fully investigate this, but we will add further discussion as it is certainly an interesting future direction.

---

> > ### Comment · Reviewer_EbFb · 2025-04-03
> >
> > Thanks for the reply. I hope the authors will update the promised changes in the new manuscript. I have also checked the comments of other reviewers and I decided to keep my score, which leans towards acceptance.

---

### Official Review · Reviewer_U4V6 · 2025-03-11

**Overall Recommendation:** 4

**Summary:**

They propose a semi-amortized approach to Bayesian experimental design, in which a policy is learned offline (as in the standard fully amoritzed approach) and then is adapted online. This increases computational cost but results in more adaptive designs, which are more robust to model misspecification. The paper is extremely well written and the method is elegant and effective.

**Claims And Evidence:**

They derive a new algorithm and claim it results in improved information gain for a given number of experiments/ samples.
They empirically demonstrate this is true on two different datasets, Source Location Finding and Constant Elasticity of Substitution (CES).
They do careful ablation studies to show where the gains come from.

**Essential References Not Discussed:**

NA

**Experimental Designs Or Analyses:**

Very solid.

**Methods And Evaluation Criteria:**

Yes

**Other Comments Or Suggestions:**

NA

**Other Strengths And Weaknesses:**

NA

**Questions For Authors:**

- For the eval metric, have you considered using synthetic data where
the ground truth theta* is available, and then asseessing distance between
E[theta|hT) and theta* where hT is from a particular design policy?

- sec 6.3. How do you evaluate diffrent designs if you just have access
to an empirical dataset, and not a DGP (simulator)?

**Relation To Broader Scientific Literature:**

Authors propose a novel and useful variant of amortized BED in which they allow the agent to learn an amortized policy offline in the usual way, and then adapt it at run time in light of observed data, by sampling potential outcomes from the posterior predictive rather than the prior predictive. This is a very elegant and useful idea.

**Theoretical Claims:**

This is an algorithms paper, so does not have new theory.

---

> ### Author Rebuttal · Authors · 2025-03-31
>
> Thank you kindly for your helpful review.
>
> > For the eval metric, have you considered using synthetic data where the ground truth theta* is available, and then assessing distance between E[theta|hT) and theta* where hT is from a particular design policy?
>
> Yes! Assessing the distance between the posterior mean and the true parameter value is indeed a reasonable metric, and one we did consider. We ultimately chose to focus on using the EIG to assess performance as this is the metric most commonly used in the BED literature and it tends to be more robust than this distance (in particular, it is possible for the posterior mean to very close to theta* while still having significant posterior variance). Preliminary results with this metric were very similar to that of the EIG and we can add such comparisons if you think they are important.
>
>
>
> >How do you evaluate different designs if you just have access to an empirical dataset, and not a DGP (simulator)?
>
> In general, this is very difficult as we are trying to evaluate the quality of a data gathering process, which cannot be directly done simply by having access to an existing dataset. In the scenario where we have some existing data and then want to gather more data, the most natural approach is to train a model to the data we already have, then use this model within an experimental design framework. Depending on context, problems like this can also sometimes be tackled using active learning or reinforcement learning methods, instead of experimental design ones.

---

### Official Review · Reviewer_2XqQ · 2025-03-14

**Overall Recommendation:** 4

**Summary:**

The paper deals with Bayesian adaptive experimental design for identifying model parameters. Fully adaptive strategies are costly and myopic. Recent work has proposed amortized experimental design in which a neural net maps from observed data directly to the experimental design policy. That work, however, is not sufficiently adaptive as it cannot learn directly from the data in the current experiment. The paper introduces Step-DAD which attempts to blend the two. Essentially it is adopts the amortized neural network approach, but adds in fine tuning based on the results of the current experiment. This improves performance relative to the amortized approach and various static approaches.

## update after rebuttal
increased score

**Claims And Evidence:**

Generally the paper is well written and the claims are supported by clear and convincing evidence. There was one claim that I did not see well-supported. The paper claims that the per-iteration cost of the fully adaptive methods is too time-consuming, and that the proposed Step-DAD is ideal for the "many problems where we can afford to perform some test-time training during the experiment itself."

However, nowhere does the paper actually give results for how much test-time training is required by the method. This is given only in terms of number-of-fine-tuning iterations, as opposed to wall time (what actually matters for the framing of the problem). I expected to see a plot showing the trade-off between achieved EIG and fine-tuning wall-time, where with 0 fine-tuning wall time we would match the EIG of DAD, and then we could see how much wall time is required to improve significantly over that, and if indeed there are many problems where we can afford that much test-time training.

If this plot were to be added, I'd be more supportive of the paper; I think its absence is significant.

**Essential References Not Discussed:**

Not aware of any.

**Experimental Designs Or Analyses:**

The evaluation was all reasonable to me.

**Methods And Evaluation Criteria:**

I have no concerns with the selection of baselines or problems.

**Other Comments Or Suggestions:**

N/A

**Other Strengths And Weaknesses:**

The paper is well-written and everything is well motivated.

**Questions For Authors:**

What does the EIG vs. wall time trade-off look like as we increase the amount of fine tuning?

**Relation To Broader Scientific Literature:**

The framing with respect to past work was well described.

**Theoretical Claims:**

Yes, Prop 3.1

---

> ### Author Rebuttal · Authors · 2025-03-31
>
> Thank you kindly for your helpful review.
>
>
> >I expected to see a plot showing the trade-off between achieved EIG and fine-tuning wall-time
>
> Thank you for the great suggestion. We have now implemented this analysis, and the chart illustrating this trade-off can be found here: https://tinyurl.com/EIG-wall-time (anonymous). We will incorporate these results into the final revised version of the paper.
>
> The results show that significant improvements can be achieved with only a few minutes of fine-tuning time, with further tuning providing additional benefits but with diminishing returns. By contrast, traditional BED approaches often require test-time computation on the order of hours (Foster et al., 2021) and similarly training the original DAD network (50k steps) also was on the order of hours.
>
>
>
> > There was one claim that I did not see well-supported. The paper claims that the per-iteration cost of the fully adaptive methods is too time-consuming, and that the proposed Step-DAD is ideal for the "many problems where we can afford to perform some test-time training during the experiment itself."
>
> We would like to emphasize that the main benefit we expect from StepDAD over the traditional greedy approach is in the quality of the designs, rather than simply in terms of cost, as is demonstrated in our numerical results. While the above results show that there are also computational cost benefits as well, our primary motivation is still to give the best possible design performance when there is computational time available during the experiment. We will make edits to the paper to ensure this is clear.
>
>
> *Adam Foster, Desi R Ivanova, Ilyas Malik, and Tom Rainforth. Deep adaptive design: Amortizing sequential bayesian experimental design. Proceedings of the 38th International Conference on Machine Learning (ICML), PMLR 139, 2021*

---

> > ### Comment · Reviewer_2XqQ · 2025-04-04
> >
> > New result looks great, thanks!

---

### Official Review · Reviewer_W51C · 2025-03-15

**Overall Recommendation:** 4

**Summary:**

This paper introduces Step-DAD, a hybrid between traditional and fully amortized policy-based approaches to Bayesian Experimental Design (DAD) that retrains its policy on-line as it gathers additional observations. The authors argue that this allows to retain the benefits of policy-based BED while overcoming its key limitation of not being able to adapt the policy in response to data collected. They discuss training procedures (mostly based on existing technical contributions). Finally, Step-DAD is evaluated in a number of experiments against DAD and other baselines, demonstrating the expected performance improvements in terms of EIG improvements and robustness to model misspecification.

## Update after rebuttal
My assessment remains unchanged - this is a good and well-executed paper and deserves to be accepted, but for me to champion the paper with a strong accept I would want to see some more novel ideas or some harder technical challenges solved.

**Claims And Evidence:**

Claims are clear and supported by convincing evidence.

**Essential References Not Discussed:**

N/A (I have limited overview of the relevant literature)

**Experimental Designs Or Analyses:**

* The analyses all appear proper and valid.
* The authors use a conservative estimate of the improvement in performance from Step-DAD over DAD, which lends additional credibility to the results.

**Methods And Evaluation Criteria:**

* The methods is a very natural evolution of DAD, and largely relies on existing methods and results used in a novel way.
* The evaluation criteria make sense, and the empirical evaluation is relatively comprehensive and includes relevant and insightful ablations.

**Other Comments Or Suggestions:**

* It would be interesting to further explore the notion of a "compute arbitrage" between investing into training a high quality policy offline vs. using a simpler policy network and investing less compute upfront and instead recover the loss in performance from that by re-training the policy online. Some of this is contained in the results of Fig 2, though that only considers the number of training steps, not the complexity of the policy network.
* In Section 6.1, you note that "the performance advantage of Step-DAD over DAD appears to be most pronounced when fine-tuning occurs just past the midpoint of the experiment, that is for
τ = 6,7 or 8.". At this point this is a bit of reading the tea leaves - this is much clearer from the Hyperbolic Temporal Discounting example in Sec 6.3.

**Other Strengths And Weaknesses:**

* The proposed Step-DAD method is a natural evolution from traditional and fully amortized approaches. The empirical results convincingly demonstrate the benefits Step-DAD - while those are intuitive and expected, there is still substantial value in confirming this in a well-designed evaluation.
* My main question that remains unanswered is how exactly how feasible this re-training of the policy is in practical settings.
  * The examples provided don't have a notion of cost or duration of obtaining a measurement, and while the studies do evaluate different and multiple steps at which to re-train the policy, this seems somewhat artificial. Why not re-train the policy at each step? It would be highly illustrative and helpful to have some real-world examples with specific budget and time considerations to understand how Step-DAD would be used in practice.
  * From a practical perspective, it appears substantially simpler to deploy a trained (and possibly inference-optimize) policy network to perform BED - e.g. this could easily be done on edge or embedded devices. It seems less straightforward to run a full training setup for re-training the policy network in such a setting. This doesn't make it infeasible or devalues the methodology more generally, but I feel these practical aspects should be discussed as well.

**Questions For Authors:**

N/A

**Relation To Broader Scientific Literature:**

* Combining the "best of both worlds" from traditional and policy-based BED where "online" computation during the experiment is permissible is a very natural thing to do and makes a lot of sense.
* The idea isn't exactly groundbreaking given the parallels with Step-Static policies or MPC (see below), but is executed solidly in the paper.
* As this extension didn't require a lot of new technical innovations, the primary contribution of the work is therefore to set up the problem and demonstrate the performance in the empirical evaluation, which was executed well.
* The Step-DAD setup is reminiscent of Model Predictive Control (MPC), which in a similar fashion re-optimizes a sequence of control inputs for the remaining experimental horizon based on a model of the process and the observations obtained so far. The main difference is that wile MPC optimizes input trajectories, Step-DAD optimizes a full policy - in that sense, MPC is the control-theory analogue of the Step-Static policy. This approach is sufficiently related that this connection warrants discussion in the paper.

**Theoretical Claims:**

Yes, Prop 3.1 (the only claim) is straightforward.

---

> ### Author Rebuttal · Authors · 2025-03-31
>
> Thank you kindly for your helpful review.
>
> > The Step-DAD setup is reminiscent of Model Predictive Control (MPC)... This approach is sufficiently related that this connection warrants discussion in the paper.
>
> Thank you for drawing this perceptive connection and we agree there are nice analogues between the approaches examined here and the Model Predictive Control literature. We will happily add references and discussion on this to the paper.
>
>
> >My main question that remains unanswered is how feasible this re-training of the policy is in practical settings
>
> Thank you for raising this important and practical point. We will add results on precise wall-clock times and additional discussion on when we expect the retraining to be feasible, including some real-world example applications and practical issues with, e.g., running on embedded devices. In short, we find that helpful retuning can usually be achieved in as little as a minute, so the retraining can be done whenever such time can be justified between one or more experiment iterations.
>
> To characterize things more precisely, we have conducted a new ablation of performance vs amount of time spent to refine the policy, the results of which can be found here: https://tinyurl.com/EIG-wall-time (anonymous). The charts demonstrate all computation is on the order of minutes and there is an initial increase in EIG with increasing wall time before a plateauing. Note that we implemented importance sampling and the varying percentage of inference corresponds to the number of samples drawn from the proposal (100% = 20,000 samples). For each row, the initial inference cost is a fixed cost and increasing finetuning steps is what increases the wall time.
>
> >Why not re-train the policy at each step?
>
> There is absolutely nothing wrong with this if sufficient computational budget is available and one may well often do so in practice. The main reasons we did not do this in our experiments were a) to keep evaluation costs down (noting that we are running many seeds over lots of different possible true theta, which won’t be needed when deploying StepDAD in practice) and b) we tended to see quite quickly diminishing returns for conducting multiple retrainings in practice.
>
> To more explicitly demonstrate these diminishing returns in the paper itself, we have run a new ablation for the location finding experiment where we increase the number of times that the policy is refined. The results, which can be found here: https://tinyurl.com/eig-interventions (anonymous), demonstrate the diminishing returns behaviour, with Total EIG roughly flat beyond 2 intervention steps. Once again note that the varying percentage of inference corresponds to the number of samples generated from the importance sampling proposal (100% = 20,000 samples).
>
> > It would be interesting to further explore the notion of a "compute arbitrage" between investing into training a high quality policy offline vs. using a simpler policy network and investing less compute upfront and instead recover the loss in performance from that by re-training the policy online. Some of this is contained in the results of Fig 2, though that only considers the number of training steps, not the complexity of the policy network.
>
> Thank you for the insightful suggestion. We hope that the new plots above provide further insight into this compute arbitrage, beyond what was already provided in Figure 2. Further investigating this trade-off with simpler policy networks instead of fewer training steps would certainly also be interesting and we will look into adding this. One important thing to note though, is that a noticeable part of the time to refine the network is in the inference rather than the network updates itself, so there will be limits to the gains which can be achieved from just simplifying the policy network itself.

---

> > ### Comment · Reviewer_W51C · 2025-04-04
> >
> > Thanks for the additional ablation results, these are quite helpful. I will keep my score - this is a good and well-executed paper and deserves to be accepted, but for a score of 5 I would have wanted to see some more novel ideas or some harder technical challenges solved.

---

### Decision · Program_Chairs · 2025-05-01

**Decision:**

Accept (poster)

**Comment:**

This paper proposes Step-DAD, a semi-amortized approach to Bayesian experimental design (BED) that extends the existing Deep Adaptive Design (DAD) framework. While DAD pre-trains a fixed policy network before experimentation, Step-DAD allows for test-time adaptation of this policy during deployment, periodically refining it based on accumulated experimental data.  The reviewers largely agree that the paper's main claim, that Step-DAD outperforms existing BED methods, is well-supported by empirical evidence across multiple experimental settings, which are commonly used in BED literature.  Reviewer 2XqQ raised a concern that the paper proposes a more efficient method, but does not present wall-clock time for its experiments.  The authors have since provided the requested analysis, satisfying that reviewer's primary concern.  While the paper lacks an overt champion, all reviewers agree that the ideas are novel and worthy of publication.  I therefore recommend acceptance based on the clear consensus among reviewers.